# Possible Role of Cannabis in the Management of Neuroinflammation in Patients with Post-COVID Condition

**DOI:** 10.3390/ijms25073805

**Published:** 2024-03-29

**Authors:** Noemi Cárdenas-Rodríguez, Iván Ignacio-Mejía, Jose Correa-Basurto, Humberto Carrasco-Vargas, Marco Antonio Vargas-Hernández, Exal Manuel Albores-Méndez, Rodolfo David Mayen-Quinto, Reynita De La Paz-Valente, Cindy Bandala

**Affiliations:** 1Laboratorio de Neurociencias, Instituto Nacional de Pediatría, Mexico City 04530, Mexico; noemicr2001@yahoo.com.mx; 2Laboratorio de Medicina Traslacional, Escuela Militar de Graduados de Sanidad, UDEFA, Mexico City 11200, Mexico; ivanignacio402@gmail.com; 3Laboratorio de Diseño y Desarrollo de Nuevos Fármacos e Innovación Biotecnológica, Escuela Superior de Medicina, Instituto Politécnico Nacional, Mexico City 11340, Mexico; corrjose@gmail.com; 4Dirección de la Escuela Militar de Graduados de Sanidad, UDEFA, Mexico City 11200, Mexico; hcarrascovargas@gmail.com; 5Subdirección de Investigación, Escuela Militar de Graduados en Sanidad, UDEFA, Mexico City 11200, Mexico; mavh78@yahoo.com.mx (M.A.V.-H.); albores_09@hotmail.com (E.M.A.-M.); 6Laboratorio de Biología Molecular, Escuela Militar de Graduados en Sanidad, Mexico City 11200, Mexico; david.invest.emgs@gmail.com; 7Laboratorio de Medicina Traslacional Aplicada a Neurociencias, Enfermedades Crónicas y Emergentes, Escuela superior de Medicina, Instituto Politécnico Nacional, Mexico City 11340, Mexico; reynadpv@gmail.com

**Keywords:** cannabis, antioxidant, neuroinflammation, post-COVID-19 condition

## Abstract

The post-COVID condition (PCC) is a pathology stemming from COVID-19, and studying its pathophysiology, diagnosis, and treatment is crucial. Neuroinflammation causes the most common manifestations of this disease including headaches, fatigue, insomnia, depression, anxiety, among others. Currently, there are no specific management proposals; however, given that the inflammatory component involves cytokines and free radicals, these conditions must be treated to reduce the current symptoms and provide neuroprotection to reduce the risk of a long-term neurodegenerative disease. It has been shown that cannabis has compounds with immunomodulatory and antioxidant functions in other pathologies. Therefore, exploring this approach could provide a viable therapeutic option for PCC, which is the purpose of this review. This review involved an exhaustive search in specialized databases including PubMed, PubChem, ProQuest, EBSCO, Scopus, Science Direct, Web of Science, and Clinical Trials. Phytocannabinoids, including cannabidiol (CBD), cannabigerol (CBG), and Delta-9-tetrahydrocannabinol (THC), exhibit significant antioxidative and anti-inflammatory properties and have been shown to be an effective treatment for neuroinflammatory conditions. These compounds could be promising adjuvants for PCC alone or in combination with other antioxidants or therapies. PCC presents significant challenges to neurological health, and neuroinflammation and oxidative stress play central roles in its pathogenesis. Antioxidant therapy and cannabinoid-based approaches represent promising areas of research and treatment for mitigating adverse effects, but further studies are needed.

## 1. Introduction

In the medical field, multiple terms are recognized for the persistent symptoms following COVID-19 infection, including post-COVID-19 syndrome, chronic COVID-19, long-term effects of COVID-19, prolonged COVID-19, long COVID-19, post-COVID-19 symptoms, persistent COVID-19 symptoms, sequelae of COVID-19, and sequelae of severe acute respiratory syndrome coronavirus 2 (SARS-CoV-2) infection, among other similar terms reflecting SARS-CoV-2 infection sequelae [1,2]. These definitions evolved from the initial conceptualization by Greenhalgh et al. who identified post-COVID-19 as a condition extending beyond three weeks after the onset of acute symptoms [3]. The symptoms, often cyclical, progressive, or multiphasic, typically persist for 2 to 4 months post-infection and have led the WHO to assign a specific classification in the ICD-10 (U09.9), referring to “Post-COVID conditions (PCC), unspecified” [2]. More than 10% of COVID-19 patients experience prolonged symptoms, affecting various body systems such as respiratory, cardiovascular, and neurological [1]. The most common symptoms include dyspnea, fatigue, olfactory and gustatory alterations among others, which may persist or fluctuate after the acute phase of the disease [2,4]. It is important to note that the PCC is not limited to severely affected patients; it is also observed in mild to moderate cases. From a pathophysiological perspective, PCC is complex and involves multiple factors, including immune and nervous dysregulation, cellular damage due to alterations in the angiotensin-converting enzyme 2 (ACE2) pathway, and inflammatory responses. The exact mechanism underlying these prolonged symptoms remains unclear, encompassing hypotheses ranging from oxidative stress to immunological alterations and inflammatory damage [1]. This review underscores the complex and multifaceted nature of neurological manifestations in the PCC, providing a deeper discussion on neuroinflammation and oxidative stress in this disease and highlighting the need for additional research to better understand and treat this condition. SARS-CoV-2 can invade the central nervous system (CNS) by binding to ACE2 receptors found in pulmonary epithelial and blood–brain barrier (BBB) endothelial cells. This invasion results in neuroinflammation, neuronal demyelination, cellular apoptosis, and coagulopathies, all of which contributed to hypoxic-ischemic neuronal injuries and BBB dysfunction. Hyperinflammation and oxidative stress are key factors in PCC’s pathophysiology. In cases of prolonged COVID-19, oxidative stress arises from mitochondrial dysregulation and NADPH oxidase (NOX) activation.

This leads to an excessive production of reactive oxygen species (ROS) and reactive nitrogen species (RNS) [5]. Inflammatory responses in the brain, triggered by the release of cytokines, increase mitochondrial ROS production [6]. Natural antioxidant systems play a crucial role in the elimination of ROS/reactive nitrogen species (RNS), and the activation of astrocytes and microglia contributes to neuroinflammation and neuronal death [7,8]. Patients with PCC experience a wide range of neuropsychiatric symptoms, including cognitive impairment, mood disorders, and various somatic symptoms. Studies suggest that these symptoms may be influenced by factors like decreased oxygen saturation [9] and increased body temperature experienced during the disease’s acute phase [10]. Post-viral somatic and mental affective symptoms have an inflammatory origin and are partially mediated by neuro-oxidative damage and reduced antioxidant defenses [11]. Antioxidant therapy has emerged as a promising strategy for mitigating disease progression; however, the low bioavailability and instability of many antioxidants limit their clinical application [7]. Specific phytocannabinoids from *Cannabis sativa*, including cannabidiol (CBD), cannabigerol (CBG), and delta-9-tetrahydrocannabinol (THC), have demonstrated potent antioxidant and anti-inflammatory effects, which could be beneficial in reducing neuronal inflammation. Given their properties, these compounds show promise in mitigating the neuroinflammation and oxidative stress associated with COVID-19, by inhibiting common amplifiers of disease progression [12] and could be effective for treating neuroinflammation in the PCC. However, it has been described that different cannabis derivatives may have conformational flexibility that could modify their affinity and even their biological effect [13].

## 2. Materials and Methods

Advanced searches were performed in PubMed, ProQuest, EBSCO, Scopus, Science Direct, Google Scholar, Web of Science, PubChem, ChemDoodle 2D server, NCBI Bookshelf, DrugBank, and Clinical Trials. We considered the original manuscripts, reviews, minireviews, systematic reviews, meta-analyses, clinical assays, books, and specialized databases. The search was conducted by applying the following keywords alone or in combination, “antioxidant”, “neuroinflammation”, “Long COVID“, “Post COVID Condition”, “chemical compounds”, “cannabis”, “chemical structure”, “anti-inflammatory”, “neuronal pathologies”, “nonneuronal pathologies”, “physiological functions”, “drug repositioning”, “neuromodulator”, “free radicals”, “reactive oxygen species”, “oxidative stresses”, “antioxidant enzymes”, “efficacy”, and “secondary effects”, as well as “Phytocannabinoids”, “Cannabidiol”, “Cannabigerol”, “Delta-9-tetrahydrocannabinol” “*Cannabis sativa*”. A total of 166 references were included.

## 3. Generalities of the PCC

The definition of PCC, characterized by symptoms persisting for more than 3 months post-COVID-19 onset, was initially introduced by the National Institute for Health and Care Excellence (NICE), the Scottish Intercollegiate Guidelines Network (SIGN), and the Royal College of General Practitioners (RCGP) [14]. This definition has been modified over the last few years by different organizations and authors. Another definition considers the moment at which symptoms appear and/or resolve and the nature of the symptoms. In general, PCC is defined as the presence of persistent post-COVID-19 symptoms following recovery from the acute phase of SARS-CoV-2 infection and consists of two stages: (1) post-acute sequelae of SARS-CoV-2 infection or post-acute COVID-19 (from week 5 to week 12 after the onset of the symptoms) and, (2) chronic post-COVID (lasting more than 12 weeks afterwards) [15]. NICE has proposed the following definitions to name the different symptomatic phases of SARS-CoV-2 infection. Acute COVID-19 usually lasts up to 4 weeks after the patient becomes symptomatic. When symptoms persist for more than 4 weeks or if late or long-term complications appear, the disease is called post-acute COVID-19 (PAC). This term encompasses both ‘persistent COVID-19’ (long COVID) and ‘post-COVID-19 sequelae’ [16]. Through a Delphi consensus led by the WHO in 2022, the authors proposed the term post-COVID-19 condition (PCC). PCC occurs in individuals with a history of probable or confirmed SARS-CoV-2 infection, generally 3 months after the onset of COVID-19, with symptoms that last at least 2 months and cannot be explained by an alternative diagnosis [2,17,18].

A wide variety of COVID-19 symptoms that have been reported in different studies, some of which include fatigue (58–80%), dyspnea (24–36%), ageusia (14–33%), anosmia (12–32%), post-activity polypnea (21–73%), cough (7–34%), memory loss (16–51%), headache (53%), depression (3–23%), and anxiety (3–26%) [19,20]. The most frequently reported symptoms are fatigue and dyspnea [19,21]. The WHO reports that approximately 10% of COVID-19 survivors may develop PCC [14]. According to surveys carried out by the United Kingdom’s Office for National Statistics, the prevalence of PCC was estimated to be between 3% and 11.7%. It is estimated that among 2 million individuals with PCC, 72% experience symptom remission within two weeks, 42% within one year, and 19% after two years [22]. The risk factors for developing PCC have not been accurately defined; according to some studies, PCC is usually more common in patients who present with moderate to severe acute COVID-19. However, PCC cases have also been reported in patients who experienced mild acute COVID-19 [13]. It has been reported that persistent symptoms are more prevalent in women and that the risk of persistent symptoms linearly increases with age [17]. The pathophysiology of symptoms due to PCC can be explained by different mechanisms depending on the specific system or organ affected. Symptoms of cognitive impairment, memory and anosmia, chronic hyperinflammation, described as secondary effects of acute SARS-CoV-2 infection, lead to the release of proinflammatory interleukins (IL-6 and IL-1), tumor necrosis factor-α (TNF-α), and ROS, which can cause long-term damage to the BBB, favoring the entry of SARS-CoV-2 into the brain, in addition to its direct entry through the nasal cavity or bloodstream, which originates the activation of microglia and favors the chain of inflammation and structural damage [13,23]. Furthermore, local microthrombosis, resulting from hypercoagulation or mitochondrial insufficiency, has also been suggested to contribute to symptoms [24]. The same structural alterations in the brain, observed in symptomatic patients who have recovered from COVID-19, have been substantiated through magnetic resonance imaging findings (diffusion tension images and three-dimensional T1-weighted sequences and pseudocontinuous arterial spin labeling) where structural modifications stand out in several brain regions, including the hippocampus, insular lobe, and olfactory cortex [25,26]. The diagnosis of PCC includes clinical signs and symptoms, and the NICE guidelines suggested a multidisciplinary approach to identify, refer, and treat these patients [16,27]. For this purpose, magnetic resonance imaging (MRI) could be particularly useful in evaluating neurological sequelae since it can detect neurodegenerative and prothrombotic changes. Regarding management, a wide range of possible treatment options have been proposed that cover different pathophysiological mechanisms, including improvement of natural killer cell function, elimination of autoantibodies, immunosuppressants, antivirals, antioxidants, mitochondrial support, and mitochondrial energy generation [28].

## 4. Neuroinflammation and Oxidative Stress in the PCC

Studies have shown that SARS-CoV-2 is neurotropic and neuroinvasive, disrupting the BBB to access brain tissue and initiating an inflammatory process that includes microglial activation, oxidative stress due to hypoxia, hypercoagulation, thrombosis, microbiome dysbiosis, accumulation of misfolded proteins, and neurological autoimmune responses which are considered principal events that induce mild neurological symptoms and headache and loss of smell to the most serious symptoms, such as stroke, encephalitis, infarctions, hemorrhagic lesions, and a hypercoagulable state [29,30,31,32]. Neurological manifestations are major complications of PCC, affecting one-third of patients, with symptoms persisting for more than a month post-COVID-19 infection [33]. According to the above findings, the patients with PCC exhibited a series of symptoms related to reactive gliosis and neuroinflammation in the CNS, including neuropsychiatric manifestations such as fatigue, ‘brain fog’ (characterized by short-term memory loss and difficulty concentrating), headache, drowsiness, cognitive impairment, sleep disorders, mood changes, disorders of smell or taste, myalgias, anxiety, posttraumatic stress, depression, sleep disturbances, sensorimotor deficits (neuropathy, paresthesia, weakness, and myalgia), and dysautonomia [1,2,34,35,36,37,38]. Among the PCC symptoms, prolonged cognitive dysfunction is one of the most common impairments affecting between 17 and 28% of the individuals, and in some cases, persisting for several years. Cognitive dysfunction, commonly observed in these patients, often includes memory impairment, attention deficit, and executive dysfunction where research has revealed a significant reduction in gray matter volume, impaired hippocampal neurogenesis, a decrease in oligodendrocytes and myelin loss causing a reduction in global brain size [39,40,41]. Chronic fatigue syndrome also recurs in patients with PCC and is characterized by immunological, neurological, and gastrointestinal alterations similar to Epstein–Barr virus (EBV)-acquired immunodeficiency in myalgic encephalomyelitis, suggesting the possibility that this virus is related to PCC [42,43,44,45].

Continuous neuroimmunological processes and oxidative stress predominantly contribute to the progression of neurological symptoms in PCC [2]. SARS-CoV-2 affects the CNS and peripheral nervous system (PNS) mainly through hematogenous or transsynaptic pathways by exploiting the ACE2 receptor present in neurons, astrocytes, endothelial cells, and muscle cells. In patients with prolonged COVID-19, a pattern of cerebral hypometabolism is observed, with implications in ACE2-rich brain regions, such as the olfactory bulb, amygdala, hippocampus, and brainstem. These hypometabolic areas, detected through positron emission tomography (PET), are associated with symptoms such as hyposmia, mood disorders, cognitive impairment, and dysautonomia. This suggests a relationship with oxidative stress, mitochondrial dysfunction, and neuronal degeneration, consistent with PCC processes [46,47]. 

Activation of the innate immune system in response to SARS-CoV-2 is known to increase cytokines, chemokines, and free radicals, affecting the BBB. This alteration permits the infiltration of immune/inflammatory cells into the CNS, activating resident immune cells such as microglia and astrocytes. Although the exact underlying mechanisms remain unclear, the neuroinflammation induced by SARS-CoV-2 and its long-term impact on the brain are currently key focuses of research [48]. It is known that the neurovascular unit (NVU), composed of neurons, microglia, and astrocytes, is key in neuroinflammation, and is primarily mediated by cytokines, chemokines, and free radicals. The NVU is crucial for communication between the immune system and the brain. Aberrant communication facilitates chronic or uncontrolled neuroinflammation, leading to the recruitment of immune cells and oxidative stress, thereby damaging brain tissue [48,49,50]. SARS-CoV-2 infection disrupts the physiological function of the NVU. Inflammatory responses in the brain have been observed, including the release of proinflammatory mediators such as IL-1β, -6 and -10, and TNF-α. These mediators can compromise the integrity of the BBB, allowing the entry of viruses and cytokines into the CNS and activating brain immune cells, resulting in neuroinflammation [48,51,52,53]. A randomized clinical trial showed that, among outpatients with SARS-CoV-2, increased IL-6 levels at the time of infection are associated with an increased risk of PCC development [54]. Postmortem and clinical studies have demonstrated microglial activation and infiltration in the brainstem and cerebellum of COVID-19 patients, accompanied by ischemic lesions and other indicators of neuroinflammation [23,49,55,56]. In addition, SARS-CoV-2 activates interferons (IFNs). A study indicated that an altered IFN-I response and chronic inflammation are associated with the persistence of symptoms in PCC. The persistent increase in IFN suggests chronic inflammation in patients with PCC [57].

SARS-CoV-2 infection has been linked to oxidative stress and neuroinflammation; in this context, the production of ROS and RNS plays a crucial role in both the elimination of infected cells and cellular signaling [58,59]. These reactive species activate immune responses, particularly by inducing the IFN response and expressing inducible nitric oxide synthase (iNOS). These pathways have been shown to significantly contribute to the innate immune response against SARS-CoV-2. Oxidative stress, resulting from an uncontrolled inflammatory response, contributes to the development of PCC [59]. Symptoms of prolonged COVID-19 suggest sustained damage to the nervous system, likely attributable to neuroinflammation and oxidative stress caused by ROS overproduction. Neuroinflammation and oxidative stress are considered the dominant pathophysiological mechanisms in PCC (Figure 1). 

It has been shown that both COVID-19 and PCC may be risk factors for long-term neurodegeneration, firstly due to chronic neuroinflammation of the neurovascular unit as we describe in Figure 2, but it may also be due to the fact that the SARS-CoV-2 nucleocapsid protein (N-protein) considerably speeds up the α-synuclein aggregation process formation that leads to the formation of multiprotein complexes and eventually amyloid fibrils, which could increase the risk of triggering Parkinson’s and Alzheimer’s disease [60]. However, more clinical studies are still needed to corroborate this in humans.

Regarding their antioxidant properties, both natural and synthetic products are being used or proposed for treating neuroinflammation in PCC patients. Natural and semisynthetic flavonoids, known for their anti-SARS-CoV-2 properties, are proposed as potential treatments for PCC [61]. Vitamin D, a potent immunomodulatory hormone with anti-inflammatory and antioxidant properties, is suggested as a potential compound for mitigating the effects of PCC due to its neuroprotective effects on modulating monoamine neurotransmission, BBB integrity, and vasculo-metabolic functions during SARS-CoV-2 infection [62,63]. A retrospective analysis performed by a group of practitioners evaluated the supplementation with PEALUT (palmitoylethanolamide co-ultramicronized with luteolin) in PCC patients and revealed that PEALUT helped in some neurological symptoms as pain, anxiety, depression, fatigue, loss of smell, and dysgeusia [64,65,66]. An eight-week supplementation with PEALUT in PCC patients restored GABAB activity and cortical plasticity, as evidenced by increased long-interval intracortical inhibition and enhanced long-term potentiation-like cortical plasticity [67]. The use of oxaloacetate, a nutritional supplement, significantly reduced fatigue in 46.8% of PCC patients when it was administered for 6 weeks [68]. Additionally, the use of Ginkgo biloba extract, delivered via nanoparticles, has been suggested for PCC patients due to its neurotherapeutic effects in various oxidative stress models, including antioxidant, antiapoptotic, and anti-inflammatory properties [8]. Additionally, a phytosomal formulation of luteolin has been proposed to mitigate brain fog associated with PCC syndrome [69]. Finally, the use of nanoantioxidants, incorporating neuroprotective compounds like curcumin, quercetin, and melatonin as nanocarriers, has been proposed as a therapeutic strategy for treating neurological sequelae associated with PCC [7].

Finally, recent clinical studies are testing new anti-inflammatory therapies against COVID-19. A phase 2 open-label interventional trial showed that aerosol inhalation of exosomes derived from human adipose-derived mesenchymal stromal cells decreased IL-6 levels in patients with severe COVID-19 [70]. In an open-label clinical trial involving patients with severe COVID-19, the use of Tranilast, a potential NLRP3 inflammasome inhibitor, was found to decrease the neutrophil-to-lymphocyte ratio (NLR) and levels of IL-1 and TNF-α [71]. A randomized clinical study showed that the use of β-1,3-1,6 glucans derived from the black yeast Aureobasidium pullulans decreased IL-6 levels in COVID-19 patients [72]. IFN-β has been proposed for use in PCC patients due to its antiviral properties [73].

## 5. Role of Cannabis Compounds in Neurological and Systemic Inflammatory Diseases

Cannabis, a plant belonging to the Cannabaceae family and originally found in Central Asia, has a composition extensively studied, revealing more than 120 phytocannabinoids. Of these phytocompounds, CBD and THC are the most abundant and best studied [74]. Considerable evidence suggests that CBD, CBG, and THC (Figure 2) possess significant antioxidative and anti-inflammatory properties. However, the exact mechanisms of action of these compounds have not yet been fully described [75].

It has been demonstrated that cannabinoid compounds act on the endocannabinoid system (ECS), via type 1 (CB1) and type 2 (CB2) cannabinoid receptors, their endogenous ligands, and the enzymes responsible for their synthesis and degradation [75]. CB1 receptors are abundant in the brain and CNS, as well as in other tissues, while CB2 receptors are primarily found in immune cells and within the cardiovascular, gastrointestinal, and reproductive systems, where they are thought to regulate several important functions [76]. CB1 and CB2 receptors can be stimulated by endocannabinoids, phytocannabinoids, or synthetic cannabinoids. An important consideration is that the affinities and potencies of different cannabinoids for CB1 and CB2 receptors vary, therefore, the efficacy and safety of one cannabinoid cannot be applied to another [76]. The CB2 receptor, and to a lesser extent the CB1 receptor, have been identified as being involved in regulating the release of proinflammatory mediators and modulation of immune cells [75]. Additionally, TNF-α, IL-1β, IL-6 and interferon gamma (IFN-γ) are among the most studied proinflammatory cytokines. Experiments with animal models have observed modulation of their levels are modulated by CBD [75]. CBD is believed to exhibit a weak affinity for CB1 and CB2 receptors, exhibiting micromolar affinity, but may also act indirectly by regulating the levels of certain endocannabinoids. CBD serves as a negative allosteric modulator of the CB1 receptor, which means it can diminish the potency of CB1 receptor ligands [77]. CBD functions as an agonist of various receptors/channels including transient receptor potential vanilloid 1 and 3 (TRPV1-3, TRPA1), peroxisome proliferator-activated receptors α and γ (PPARγ), serotonin receptor (5-HT1A), and adenosine receptors A2 and A1. Additionally, it acts as an antagonist of orphan G-protein coupled receptors (GPR55, GPR18), and 5-HT3A. CBD is also an inverse agonist of receptors GPR3, GPR6, and GPR12 [78]. Moreover, activation of the CB1 receptor has been demonstrated to amplify proinflammatory signaling and increase oxidative stress [79,80]. Evidence provided by the results of several studies in animal and in vitro models of different pathologies including pulmonary, CNS and PNS, epithelial, and systemic inflammation, among others, show that blockade of CB1 receptors can produce a reduction in inflammation and oxidative stress. Moreover, the activation of CB2 receptors has been shown to reduce inflammation and oxidative stress, thereby decreasing inflammatory damage, and contributing to clinical improvement [77,81].

Regarding neuroinflammation, as induced in an animal model of pneumococcal meningitis, it was shown that chronic treatment with CBD (2.5, 5, or 10 mg/kg) significantly reduced TNF-α concentrations in the frontal cortex, preventing cognitive impairment [82]. In an in vitro inflammatory response study, the combination of CBD with *Morinda citrifolia* extract was evaluated in RAW264 cells stimulated with lipopolysaccharide (LPS). This combination was found to be more effective at inhibiting nitric oxide production and reducing iNOS expression than CBD alone. These findings suggest the therapeutic potential of CBD in the treatment of neuroinflammatory conditions and highlight its synergistic effect with other compounds in modulating inflammation [83]. Similarly, a study on orofacial dyskinesia in mice, induced with haloperidol, investigated the effects of CBD. CBD (60 mg/kg) administered daily for 21 days in the striatum attenuated microglial activation, oxidative stress, and proinflammatory cytokines (TNF-α and IL-1β) and increased the anti-inflammatory cytokine IL-10 [84]. In experimental autoimmune encephalitis, administration of CBD at 20 mg/kg daily from 9 days post disease induction until day 25 was associated with delayed onset and reduced severity of the disease, decreased T-cell infiltration and reduced levels of IL-17 and IFN-γ. Interestingly, CBD treatment led to a profound increase in myeloid-derived suppressor cells (MDSCs). These MDSCs strongly inhibit T cell proliferation [85]. In a study on neuropathic and inflammatory pain induced in the sciatic nerve, CBD at doses of 2.5, 5, 10, and 20 mg/kg significantly reduced hyperalgesia in response to mechanical and thermal stimuli. CBD also decreased prostaglandin E2, lipid peroxide, and nitric oxide levels and normalized antioxidant enzyme activity, without affecting TNF-α or NF-κB levels [86]. In a study of tardive dyskinesia, the results showed that IL-10 levels increased in the striatum of animals receiving CBD. In addition, CBD decreased LPS-induced oxidative stress and microglial activation [12]. Additionally, CBD demonstrated the neuroprotective effect on endoplasmic reticulum stress in STHdhQ7/Q7 striatal cells. CBD pretreatment increased cell viability and the gene expression of the pro-survival chaperone GRP78 and the neurotrophic factor MANF, while decreasing the expression of pro-apoptotic markers such as BIM and caspase-12, indicating that CBD may protect against endoplasmic reticulum stress-induced cell death [87]. CBD demonstrated anti-inflammatory and antinociceptive effects in conditions induced by zymosan. CBD (5 mg/kg), was shown to be more effective at reducing inflammation and pain, as evidenced by the significant decrease in TNF-α levels [88]. In a rat model of Parkinson’s disease, CBD administered at 10 mg/kg intraperitoneally for 28 days reduced nigrostriatal neurodegeneration, alleviated the neuroinflammatory response, and improved motor performance. CBD has been shown to promote the neuroprotective effect of ciliary neurotrophic factor on astrocytes through its activity on TRPV1 receptors [89]. The anti-inflammatory or antioxidant efficacy of CBD has also been reported in other neurological disease models, such as those of epilepsy [90], multiple sclerosis [91], and ischemic insult [92].

In other experimental studies, the use of CBD has been proposed to have therapeutic effects on other systemic inflammatory diseases. In cases of local and systemic inflammation, induced by croton oil and LPS, respectively, local administration of CBD was found to reduce myeloperoxidase (MPO) activity, circulating TNF-α, and the development of edema. Systemically, CBD decreased TNF-α and IL-6 levels in a dose-dependent manner, with 100 μg of topical CBD showing anti-inflammatory effects comparable to those of intraperitoneal administration [93]. In a study on periodontitis, daily administration of CBD (5 mg/kg) for 30 days significantly reduced IL-1β and TNF-α expression in gingival and bone tissues. A decrease in MPO activity was also observed, indicating a reduction in neutrophil migration and overall inflammation [94]. In ischemia/reperfusion-induced renal injury, CBD (5 mg/kg) administered directly into the aorta significantly reduced TNF-α and IL-1β levels, thereby decreasing damage caused by oxidative stress [95]. In hepatic steatosis, CBD (5 or 10 mg/kg) attenuated the effects of alcohol, as indicated by a decrease in the serum transaminase level, hepatic inflammation, and neutrophil accumulation. A reduction in oxidative/nitrative stress in the liver was also observed [96]. In a colitis model induced by 2,4,6-trinitrobenzenesulfonic acid, both CBD and O-1602, a synthetic cannabinoid, were administered 30 min before the inducing agent and continued daily for 5 days. Both CBD (1 mg/kg) and O-1602 (10 mg/kg) significantly reduced MPO and IL-6 levels, thereby indicating a reduction in inflammation and tissue damage [97]. In addition, in autoimmune type 1 diabetes, CBD treatment (5 mg/kg) significantly decreased the proinflammatory cytokines TNF-α and IFN-γ in plasma, as well as Th1 cytokine production in T cells and peritoneal macrophages, while maintaining Th2 cytokines such as IL-4 and IL-10, which decreased the incidence and delayed the onset of the disease [98]. In inflammatory bowel disease, daily administration of CBD (5 mg/kg) significantly reduced colonic tissue damage and decreased nitrite production and oxidative stress levels [99]. In acute pancreatitis, CBD (0.5 mg/kg) and O-1602 (10 mg/kg) significantly reduced MPO activity and levels of IL-6 and TNF-α in plasma and pancreatic tissue [49,100].

THC primarily interacts with CB1 and CB2 receptors with nanomolar affinity, acting as a partial agonist [25]. THC acts as an agonist of PPARα-γ receptors, GPR55, GPR18, TRPV2-4, and TRPA1 channels, and as an antagonist of the TRP cation channel subfamily M member 8 (TRPM8) and serotonin 3A receptor (5-HT3A) [74,101]. Furthermore, like CBD, THC has shown activity at the μ and δ opioid receptors [102]. THC has not been shown to reduce proinflammatory cytokine levels or to increase anti-inflammatory cytokines in models of neurological diseases, although significant improvements in neuropathic pain have been reported [75,103,104]. However, THC has recently been proposed as a new neuroprotective agent in Alzheimer’s disease due to its ability to restore cell viability in retinoic acid-differentiated neuroblastoma SH-SY5Y cells and upregulate genes involved in endoplasmic reticulum stress and the unfolded protein response [105]. Additionally, it was demonstrated that THC (5 mg) combined with codeine (50 mg) exhibited analgesic and antispastic effects in a patient with spasticity and pain due to spinal cord injury [106]. In other works, in adipose tissue under obese conditions, THC (10 mg/kg) was administered daily for 10 days increasing gluconeogenesis. Interestingly, THC also increased macrophage infiltration and the expression of TNF-α [107]. Additionally, in a retinal damage study, daily administration of THC (1 or 2 mg/kg) for two months induced a significant increase in oxidative stress, manifested by an increase in malondialdehyde (MDA) levels, a decrease in reduced glutathione (GSH), superoxide dismutase (SOD), and catalase (CAT) activity and an increase in NOX-4 protein expression. In addition, there was an increase in nitric oxide, IL-1β, IL-6, and TNF-α production in mouse retina [108]. In a study on the pancreas, the effects of THC were investigated in rats with hyperinsulinemia. The groups received THC (1.5 mg/kg) daily, which reduced the mRNA expression of IL-6, NF-κβ, and TNF-α. In addition, THC reduced the levels of total oxidative capacity [109].

CBD, when combined with THC, has demonstrated anti-inflammatory effects in several neurological disease models. In a study on experimental autoimmune encephalitis, administering a combination of CBD and THC daily (10 mg/kg, 1:1 ratio), starting 10 days after disease induction, was found to reduce the severity of encephalitis, as indicated by decreased IL-17A and increased interleukin-10 (IL-10) [110]. Furthermore, in experimental autoimmune encephalitis, combinations of THC and CBD (10:10 and 1:20; 215 mg/kg) were administered orally from day 6 to 18 post disease induction. These formulations reduced TNF-α and increased brain-derived neurotrophic factor (BDNF) [111]. In a murine model of persistent inflammatory pain, THC (2.0 mg/kg) and CBD (10 mg/kg) administered twice daily for three days showed that THC had no significant impact on serum cytokine levels [112]. Other experimental studies have demonstrated that CBD and THC suppress neuroinflammation in autoimmune encephalomyelitis by decreasing pro-inflammatory molecules and enhancing the anti-inflammatory phenotype associated with alterations in miRNA profiles and gut microbiota [110,113]. Therefore, these results suggest the potential of cannabinoids as effective treatments for diseases involving chronic inflammation, such as PCC [75].

CBG binds to both CB1 and CB2 receptors, acting as a partial agonist. CBG has been shown to be a poor ligand that binds weakly without activating these receptors. CBG is a strong agonist of α2-adrenoceptors, whereas it is only a moderate competitive antagonist of the 5-HT1 receptor. Experiments have shown that CBG inhibits COX enzymes at higher concentrations than conventional anti-inflammatory compounds [114]. CBG is known to exhibit anti-inflammatory and antioxidant properties [115]. In relation to neurological diseases, it has been observed that in models of multiple sclerosis CBG attenuated the microglial production of nitric oxide and TNF-α levels are significantly reduced, restoring the neuronal loss [116,117]. VCE-003, a derivative of CBG quinone, has also demonstrated protective activity against neurodegeneration and neuroinflammation in various experimental models of Huntington’s disease and Parkinson’s disease [118,119,120,121]. Moreover, combinations of CBD and CBG have been effective in reducing neuroinflammation in the motoneuron-like cell line NSC-34, particularly when induced by LPS; this decreasing iNOS activity and increasing nuclear factor erythroid 2-related factor 2 (Nrf-2) levels [122]. In a murine model of colitis, CBG (1–30 mg/kg) was administered for two days, starting 24 h after induction. CBG reduced proinflammatory cytokines IL-1β and IFN-γ, increased the anti-inflammatory cytokine IL-10 levels and reduced the activity of enzymes such as MPO and iNOS, without affecting COX-2 [123]. A study using 3D human skin equivalents demonstrated that both CBG and CBD effectively reduce ROS in human dermal fibroblasts, with CBG being approximately 1800 times more potent than ascorbic acid (vitamin C) in this regard. In addition, CBG inhibited the release of IL-1β, IL-6, IL-8, and TNF-α in response to inflammatory inducers such as ultraviolet light and sunlight [115].

In relation to clinical trials, the use of cannabis and its metabolites or agonists, was tested under several inflammatory conditions. In patients with progressive multiple sclerosis treated with IFN-β-1b, it was shown that the cannabis formulation Sativex^®^ (50:50 CBD and THC) decreased cannabinoid type 2 receptor expression levels in peripheral blood mononuclear cells (PBMCs) [124]. In a randomized, double-blind, placebo-controlled study, it was shown that CBD (20 mg/kg/day) attenuates symptoms in patients with idiopathic and diabetic gastroparesis, reducing gut inflammation [125]. CBD also has showed the capacity to decrease inflammation in the human gastrointestinal tract in vitro, ex vivo, and in vivo, increasing gut permeability [126]. In randomized, placebo-controlled studies, lenabasum (20 mg/day), a cannabinoid receptor type 2 agonist, was shown to effectively improve conditions such as dermatomyositis, and a combination of CBD and aspartame effectively reduced inflammation in atopic dermatitis [127,128]. The use of lenabasum (20 mg/day) also decreased IL-8 and IgG levels in cystic fibrosis and decreased inflammation and fibrosis in systemic sclerosis patients [129,130]. Moreover, in a randomized, parallel, double-blind study, it was shown that CBD formulations (30 mg) decrease IL-10 and TNF-α kevels in the PBMCs of healthy individuals [131]. Finally, CBD, CBG, and THC + CBD have shown anti-inflammatory and antioxidant properties in various pathologies involving neuroinflammation (Figure 3a,b), making these cannabis compounds potential treatments for PCC.

## 6. Possible Therapeutic Effects of Cannabis Compounds on PCC Neuroinflammation

Currently, CBD’s use as an antiviral therapy for PCC is under exploration, especially since 7-hydroxy-cannabidiol, a derivative of CBD, has been shown to inhibit SARS-CoV-2 replication by blocking viral gene expression, upregulating interferon expression, and downregulating ACE2 and transmembrane protease serine 2 (TMPRSS2) [132,133]. A randomized, placebo-controlled, single-blind, open-label crossover study has shown that a formulation containing CBD can improve symptoms in patients with PCC. In this study, two treatment groups were randomized: one group received blinded active product for 28 days (Group 1, n = 15) and the other group received a blinded placebo for the same period (Group 2, n = 16). The Patient-Reported Outcomes Measurement Information System (PROMIS^®^) scores and the Patient Global Impression of Change (PGIC) score were used to measure the efficacy of Formula C™, a cannabidiol (CBD)-rich. PROMIS^®^ is a set of patient-centered instruments used to measure various patient reported outcomes. High scores suggest more of what is being measured (e.g., more fatigue, more shortness of breath) and a lower score suggests improvement in symptoms or complaints assessed depending on the symptom or complaint being measured. PGIC evaluates various aspects of patients’ health and assesses if there has been an improvement or decline in clinical status. The scale associated with subjective responses ranges from 1 to 7, higher ratings indicate improved symptoms (responders) versus those reporting lower scores suggesting no change or a worsening of their condition (non-responders). The PROMIS^®^ questionnaires showed the beneficial effect of the Formula C™, significantly improving dyspnea, anxiety, and sleep disturbance between day 0 and day 28 in Group 1. The analysis between two groups revealed differences in the ability to participate in social roles, satisfaction with social roles and cognitive function, improved in both groups, but slightly favored Group 2. The PGIC score also showed the same differences between Group 1 and Group 2, where the ability to participate in social rules, cognitive function, and satisfaction with social rules also slightly favored Group 2 over 28 days. The authors concluded that, in persons with PCC who are nonresponsive to conventional therapies, this study demonstrated symptom improvement utilizing Formula C™. [134]. Regarding clinical trials on the use of cannabis and its metabolites for treating PCC, an open-label, phase 2 clinical trial is investigating the feasibility of using CBD as a medicinal cannabis treatment named MediCabilis Cannabis sativa 50 for various symptoms of PCC. In this assay, patients used medication daily for a total of 21 weeks (2-week titration period, 18-weeks steady dose, 1-week dose reduction), followed by 3-weeks with no medication and patients submitted a monthly self-report assessing common symptoms including breathlessness, fatigue, mood, cognition, and pain via a smartphone app, as well as real-time data on heart rate, physical activity, and sleep using wearable technology and also submitted daily self-reports of key symptoms (mood, pain, fatigue, and breathlessness) via a smartphone app for 7 days per 28 days. For the evaluation of PCC symptoms were used the Yorkshire rehabilitation scale (long-COVID symptoms), fatigue severity scale (fatigue), EuroQol 5 Dimensions (quality of life), brief pain inventory short form (pain), generalized anxiety disorder assessment (anxiety), patient health questionnaire (depression), and the Pittsburgh sleep quality index (sleep quality). Wearable technology was used to measure resting heart rate and activity levels. The current status of the study is completed [135]. A double-blind, randomized, placebo-controlled, single-center clinical trial is underway to explore the safety and efficacy of a full cannabis flower formulation, rich in cannabinoids and terpenes, named Xltran Plus™ and Xltran™ in treating PCC patients with prolonged symptoms caused by COVID-19 during 28 days. The treatment will consist of daily doses of Xltran Plus™, Xltran™, or a placebo. Xltran Plus™ contains 10.42 mg of cannabinoids, 0.55 mg tetrahydrocannabinol (THC), and 2.729 mg of terpenes per 0.25 mL of solution. Xltran™ contains 0 mg of cannabinoids, 0 mg THC and 1.28 mg terpene per 0.25 mL of solution. Participants will take 1 mL sublingually after the morning meal and 1 mL sublingually after the evening meal for 28 days. The PGIC will be used for the evaluation of symptoms (for the individual perception of overall improvement), the total symptom score (to evaluate symptoms, specifically: fatigue/weakness, pain, brain fog, dysautonomia, headaches, sensory problems, sleep difficulties, shortness of breath, flu-like symptoms, and mood disorders), the insomnia severity index (insomnia), the Harvard flourishing index (life satisfaction, mental and physical health and social relationships), DANA Brain Vital (cognitive efficiency), hours of upright activity, and daily steps. As of the time of writing, this study has not started recruiting participants [136]. Finally, a randomized parallel assignment clinical trial is investigating the effects of CBDRA60, a sublingual tablet combining Cannabidiol and Gigartina Red Algae, aimed at reducing the duration of symptoms in COVID-19 patients for the prevention of post-sequelae of the infection. The patients will receive CBDRA60 (30 mg CBD, 30 mg RA/60 mg combo; 2x/daily with food or 120 mg total) or a placebo in a 1:1 ratio. The study duration will be 5 weeks (35 days). The study will measure the decreased hospitalization and resolution of COVID-19 symptoms through to the time at which the patient is completely symptom free (fever, cough, shortness of breath, fatigue, muscle or body aches, headache, new loss of taste, new loss of smell, congestion or runny nose, nausea, vomiting, diarrhea, and shortness of breath) using a 0–3 scale where lower is better and higher is worse assessed symptoms. Similarly, this clinical trial has also yet to commence recruiting participants [137]. To date, these clinical trials have no publications or results available. Additionally, CBD has been shown to exert neuroprotective effects by attenuating brain damage related to neurodegenerative, cognitive, and/or ischemic conditions, as well as alleviating psychotic, anxiety, and depressive behaviors, thereby facilitating synaptic plasticity [33,138,139,140,141]. Regarding the therapeutic activity of cannabis in treating symptoms related to PCC neuroinflammation, there are currently only a few experimental or clinical studies that demonstrate the effects of the plant or its metabolites. In a clinical case, a 60-year-old patient exhibited persistent symptoms of muscle joint tenderness and pain, generalized fatigue, and depression, which worsened with minimal physical activity approximately two weeks after recovering from a SARS-CoV-2 infection with polyarthralgia, depression, and fatigue being the major complications [142]. The described patient had a history of mild malaise before she was infected, but she was physically active and efficient and was involved with all of her usual family activities. Her PCC changed radically to the point of rendering her too weak to even minimally manage her usual daily activities. The physical examination evidenced signs and symptoms compatible with the patient’s complaints. Interestingly, she described the tender, painful knots produced by pressure exerted on various body areas (myalgia) and joints (arthralgia). The patient rated the level of pain as 8–9/10 on a visual-analog scale (VAS) and described her mood as having had deteriorated significantly (Zung depression scale ~70). As various drugs did not alleviate her symptoms, her physician administered medical cannabis. The patient used THC/CBD cannabis several times at concentrations of 20%/4% which improved most of her symptoms for several hours and then the patient was provided with the same cannabis composition for inhalation/smoking (daily maximal dose of 15 mg/kg). After a week of use, the patient experienced significant improvement in symptoms such as pain, insomnia, and depression without adverse events. Her pain VAS decreased to 3–4, and she was now able to complete housework, go for walks, and sleep well. The patient was much less depressed (Zung depression scale 45), and the myalgia and arthralgia were diminished. The patient described its status as “getting her life back”. This improvement remained stable even three months after initiating medical cannabis treatment. The authors concluded that the use of medical cannabis, as mixture of THC/CBD, is an optimal protocol for attenuating PCC physical and mental sequelae, without evoking adverse effects due to the patient’s performance stabilizing—her sense of humor returned to normal, and pain was mitigated to a level similar to her pre-COVID-19-infected state [142]. A recent site-specific, grid-based docking study has suggested that cannabis might be effective in treating post-COVID-19 neurodegeneration. This study selected eight potential cannabis compounds, with cannabivarin (CNV) and CBD emerging as the most promising. CBD showed the best docking with ACE2 and interleukin-6 (IL-6), and CNV docked with the TMPRSS2 and neuropilin 1 (NRP1) proteins [143]. The authors concluded that cannabis could be effective for PCC treatment, targeting major post-COVID elements like ACE2, IL-6, and TMPRSS2. CBD and CNV may act on neurodegenerative problems associated with PCC [143,144]. β-Caryophyllene (BCP), acting as a cannabinoid receptor 2 agonist, is proposed to aid in COVID-19 recovery and reduce organ damage, thanks to its antioxidant and anti-inflammatory effects [145]. Another study, however, suggests that inhibiting the endocannabinoid/CB1 receptor and iNOS activity might reduce long-term health complications from COVID-19 [146]. Additionally, some studies have proposed or demonstrated the underlying mechanisms or therapeutic targets related to PCC development and where cannabis, or its metabolites, could interact. In this sense, brain leucocytes are proposed as therapeutic targets for post-COVID-19 brain fog, with patch-clamp studies showing that drugs like antihistamines, statins, and anti-inflammatory agents can suppress Kv1.3 channel activity (involved in neuroinflammation and microglial and lymphocyte hyperactivation) and reduce the production of proinflammatory cytokines [147]. Cannabis, particularly CBD, has potential for treating neurological symptoms in PCC patients due to its anti-inflammatory and immunosuppressive effects, mast cell modulation properties, and attenuation of neuropathic pain, potentially via FKBP5, a protein related to NF-kB activation [148,149,150]. The role of transient receptor potential Melastatin 3 (TRPM3), a channel associated with multiple biological processes such as cell differentiation and division, apoptosis, transcriptional events, cell adhesion, and immune synapse formation regulating Ca^2+^ signaling in CNS, has also been observed in the PCC where the activity of this channel is reduced [151]. Cannabinoid receptor activation is known to induce cell apoptosis. Specifically, CBD significantly reduces cell viability, likely through antagonizing the transient receptor potential melastatin type-8 (TRPM8). This compound could also modulate the activity of TRPM in PCC patients and exert neuroprotective effects through proapoptotic effects on SARS-CoV-2 [152]. A study found that patients with the combined GSTM1-null/GPX1LeuLeu genotype are more prone to PCC-related brain fog, while the GSTP1 Ile and GSTO1 Ala alleles together increase the risk of PCC-related myalgia [153]. If genetic susceptibility influences PCC, utilizing cannabis or its metabolites, especially for their impact on antioxidant enzyme activities (like those of CBD and THC), might improve neurological symptoms in PCC patients [154]. The S1 spike protein is thought to be responsible for PCC syndrome as it reportedly causes endothelial damage, BBB disruption, and autoimmunity against endothelial cells on its own [155,156,157,158,159,160]. Cannabinoid acids, such as cannabigerolic acid and cannabidiolic acid, have been identified as ligands with micromolar affinity for the spike protein, effectively preventing the infection of human epithelial cells [161]. This finding suggests cannabis as a potential option for blocking SARS-CoV-2 entry into neuronal cells and thereby preventing the development of PCC.

Another treatment strategy for PCC patients not related to Cannabis or its metabolites involves supervised exercise programs. For instance, a study showed that 65% of 14 women with fatigue responded positively to such a program, evidenced by improvements in walking distance and oxygen saturation with stability in the percentage of meters walked. Patients with obesity (21.4%) or double-vaccinated against SARS-CoV-2 (50%) also decreased fatigue [162]. Additionally, pacing (consisting of adapting and adjusting the different patients’ activities in terms of physical, cognitive, and emotional effort within the limits imposed by the illness) has been shown to improve various symptoms in PCC patients (including cognitive impairment, headache, dysautonomia, neurosensory disturbances, fatigue, and sleep disorders). A retrospective study showed that pacing generated recovery and improvement rates were 33.7% and 23.3%, respectively, in 86 patients with PCC. Additionally, a longitudinal cohort study of 31 participants with PCC demonstrated that a structured pacing protocol over 6 weeks significantly reduced the post-exertional symptom exacerbation episodes with an average decrease of 16% each week, and a reduction across all three exertional triggers (physical, cognitive, and emotional) [163,164]. Other therapeutic options have been proposed, including meditation, music therapy, disease-modifying therapies, ozone therapy, and the ingestion of probiotics or foods with anti-inflammatory properties [65,165,166]. These treatment options, potentially in combination with medical cannabis or its metabolites, could further improve the biological, physiological, and neurological status of PCC patients.

## 7. Conclusions

PCC is a pathology characterized by persistent symptoms following a COVID-19 infection, with neurological sequelae being among the major complications. Neuroimmunological dysfunctions and oxidative stress are considered the cause of chronic hyperinflammation in long-term COVID-19 patients. Different compounds with anti-inflammatory or antioxidant properties have been employed in treating COVID-19 patients, potentially preventing progression to PCC. Cannabis compounds, including CBD, CBG, and THC, either alone or combined, as well as receptor agonists, have been utilized in experimental models and clinical studies for the management of different types of neurological and systemic inflammatory conditions, such as encephalitis, pain, cancer, diabetes, skin conditions and renal, gastrointestinal, and hepatic injury, due to their antioxidant and anti-inflammatory properties. CBD has been proposed as an antiviral therapy against COVID-19 because of its ability to block SARS-CoV-2 replication. Moreover, in different clinical trials, the use of cannabis or CBD for the treatment of PCC-related neurological symptoms is being tested because CBD has neuroprotective effects. To date, only a few studies have explored or demonstrated the protective mechanisms of cannabis or its metabolites in PCC neuroinflammation. CBG, CBD, and TCH have therapeutic effects through their interaction with different receptors that participate in neuroprotective processes. However, we must not omit that these molecules have conformational flexibility and require study in the context of neuroinflammation. Cannabis or its metabolites, whether used alone or in combination with other antioxidants and treatment strategies, may serve as effective adjuvants in managing neurodegeneration caused by inflammatory and oxidative processes in COVID-19 patients and subsequent PCC. However, this requires validation through further experimental studies and clinical trials.

## Figures and Tables

**Figure 1 ijms-25-03805-f001:**
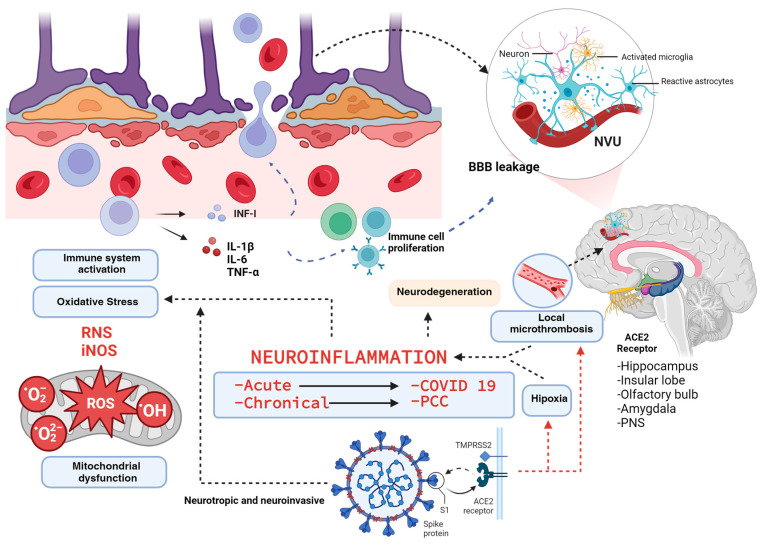
Neuroinflammation in COVID-19 and Post-COVID Condition. SARS-CoV-2, a neurotropic and neuroinvasive virus, affects both the central and peripheral nervous systems, particularly targeting regions expressing ACE2 receptors. This interaction not only activates the immune system but also leads to microthrombosis and hypoxia, exacerbating the inflammatory response through the release of cytokines such as IL-1β, IL-6, TNF-α, and INF-I. Concurrently, there is an upregulation of nitric oxide synthase (iNOS) and an increase in oxygen and nitrogen free radicals. These changes prompt the activation of the neurovascular unit (NVU) and compromise the integrity of the blood–brain barrier (BBB), facilitating the infiltration of inflammatory cells. This cascade of events amplifies neuroinflammation, which, if sustained, may lead to chronic neurodegeneration.

**Figure 2 ijms-25-03805-f002:**
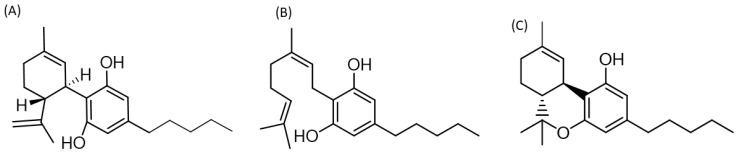
Two-dimensional structure of cannabis derivatives with possible therapeutic effects for PCC. (**A**) Cannabidiol (1′R,2′R)-5′-methyl-4-pentyl-2′-(prop-1-en-2-yl)-1′,2′,3′,4′-tetrahydro-[1,1′-biphenyl]-2,6-diol, (**B**) Cannabigerol (Z)-2-(3,7-dimethylocta-2,6-dien-1-yl)-5-pentylbenzene-1,3-diol, (**C**) Delta-9-tetrahidrocannabinol b(6aR,10aR)-6,6,9-trimethyl-3-pentyl-6a,7,8,10a-tetrahydro-6H-benzo[c]chromen-1-ol. Two-dimensional structures were built using the ChemDoodle 2D server [https://www.chemdoodle.com/] (accessed on 27 February 2024).

**Figure 3 ijms-25-03805-f003:**
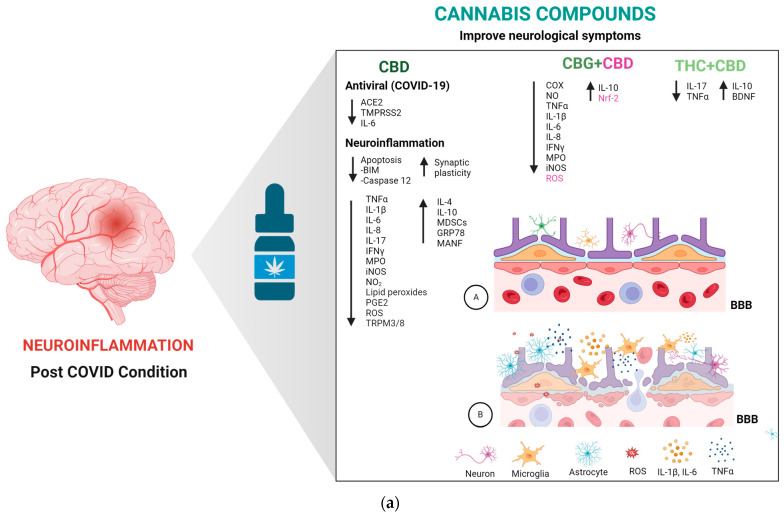
(**a**). Potential anti-inflammatory and antioxidant effects of cannabis compounds in COVID-19 and post-COVID condition (PCC). CBD (cannabidiol) alone, and in combination with CBG (cannabigerol) and THC (Delta-9-tetrahydrocannabinol), exhibits similar potential antiviral, anti-inflammatory, and antioxidant properties in the context of COVID-19. These effects could benefit patients with PCC. (**A**) Undamaged blood–brain barrier. (**B**) Neuroinflammation in PCC. (**b**) Interaction of Cannabinoids CBD, CBG, and THC with CNS cells and their impact on modulating inflammation and oxidative stress in the post-COVID condition. This figure demonstrates how cannabidiol (CBD), cannabigerol (CBG), and tetrahydrocannabinol (THC) engage with neurons, astrocytes, and microglia within the central nervous system (CNS) during the post-COVID condition to influence the production of pro-inflammatory cytokines and oxidative stress markers. By activating cannabinoid receptor types 1 (CB1) and 2 (CB2), CBD, CBG, and THC reduce the secretion of pro-inflammatory agents such as tumor necrosis factor-alpha (TNFα), interleukin-1 beta (IL-1β), interleukin-6 (IL-6), interleukin-8 (IL-8), interleukin-17 (IL-17), and interferon-gamma (IFNγ), as well as myeloperoxidase (MPO), inducible nitric oxide synthase (iNOS), nitrogen dioxide (NO2), prostaglandin E2 (PGE2), and reactive oxygen species (ROS), thus fostering a neuroprotective environment. The vertical black arrows indicate the dynamics of production for various elements: a downward orientation reflects a decrease in the production of the elements specified next to such an arrow, whereas an upward orientation indicates an increase. On the other hand, the green arrows highlight the specific effect of each phytocannabinoid upon interacting with its corresponding receptor.

## Data Availability

Not applicable.

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
