# Peer review of "Possible Role of Cannabis in the Management of Neuroinflammation in Patients with Post-COVID Condition"

_ijms, 2024, doi:10.3390/ijms25073805_

Round 1

Reviewer 1 Report

Comments and Suggestions for Authors

The article contains interesting information on the beneficial effects of cannabis in neuroinflammation that develops after COVID-19. The article is well structured and contains two excellent schemes describing both the development of inflammation and the mechanisms of possible action of cannabis and its derivatives. However, in the latter (or separate) scheme, I would have liked to see not only the ultimate effect of cannabis, but also the specific targets that cannabis interacts with. In my opinion, only stating anti-inflammatory and antioxidant effects is not enough, especially since cannabiods have specific receptors. In addition, there is specific information about the interaction of COVID-19 proteins (spike-protein and N-capsid protein) with amyloidogenic proteins, particularly alpha-synuclein. Obviously, neurodegenerative processes may be associated with such interaction and cannabiods may influence their progression. If the authors have such information or assumptions of this kind, they should be included.

Author Response

  1. The article contains interesting information on the beneficial effects of cannabis in neuroinflammation that develops after COVID-19. The article is well structured and contains two excellent schemes describing both the development of inflammation and the mechanisms of possible action of cannabis and its derivatives.

RESPONSE: We appreciate your comment as an expert.

  1. However, in the latter (or separate) scheme, I would have liked to see not only the ultimate effect of cannabis, but also the specific targets that cannabis interacts with.  In my opinion, only stating anti-inflammatory and antioxidant effects is not enough, especially since cannabinoids have specific receptors.

RESPONSE: We agree with your suggestion, we added a figure 3b explaining the main mechanisms of action of CBD, CBD+THC and CBD+CBG

  1. In addition, there is specific information about the interaction of COVID-19 proteins (spike-protein and N-capsid protein) with amyloidogenic proteins, particularly alpha-synuclein. Obviously, neurodegenerative processes may be associated with such interaction and cannabiods may influence their progression. If the authors have such information or assumptions of this kind, they should be included.

RESPONSE: thank you for your suggestion, effectively, there are relationship between alpha-synuclein and cannabinoids, the later as neuroprotectors: https://pubmed.ncbi.nlm.nih.gov/36108815/, https://pubmed.ncbi.nlm.nih.gov/36964823/ however, these studies have been carried out mainly in vitro, but they confirm the potential of cannabinoids to prevent neurodegeneration, a complication that patients with CCP and COVID-19 can present in the long term. We focus on this topic briefly in the text as you suggested.

Reviewer 2 Report

Comments and Suggestions for Authors

The review investigates the use of cannabinoids as a counter to post-COVID condition (PCC), specifically neuroinflammation, its symptoms and subsequent related neurodegenerative diseases. Here CBD, CBG and THC are highlighted as potent phytocannabinoids combat the neurodegenerative effects of PCC.

The scope is adequate and the narrative is appropriate.

The problem of PCC is introduced concisely and is categorised into a range of mechanisms: oxidative stress, immunological alterations and inflammatory damage. This reviewer sees this as a good attempt at parsing through the research on PCC given how under-developed the field is and how variable the distribution of symptoms can be.

To further include the latest research on the molecular structure of cannabinoids and strengthen the fit of the review to the scope of International Journal of Molecular Science, on the role of cannabinoids (section 5), this reviewer recommend referencing the conformational flexibility of CBG as a distinguishing factor that relates to its inhibitory character and pharmacodynamics profile: Bioactivity of the cannabigerol cannabinoid and its analogues–the role of 3-dimensional conformation. Organic & Biomolecular Chemistry, 2023.

Also, to illustrate the molecular structure of these discussed cannabinoids may be informative.

On possible alternative therapies (section 6) it would be more useful to quantify the data from each study into the review (such as line 500 “can improve” by how much?). The aim here is that at least with the values present in each mention of each study, this section could be appended with a clearer idea of an otherwise foggy direction that this branch of research has. To summarise, based on the data amassed, how likely and to what extent are these alternative therapies contributing to the reduction in PCC inflammation?

Line 156 repeated use of “can cause”

Please confirm the originality of the 2 figures. If they are not original, proper references should be given and permission to use should be granted.

A question for the authors that is maybe outside of the scope of this review, around line 535 there is the case study of a 60 yr old man using cannabinoids to alleviate his symptoms. Is there a direct/quantitative way to measure and monitor PCC outside of its secondary symptoms (pain insomnia etc). Is the relationship between curing these secondary symptoms and reduction of PCC neurological or psychological (a la placebo effect)?

Comments on the Quality of English Language

minor editing needed

Author Response

  1. The review investigates the use of cannabinoids as a counter to post-COVID condition (PCC), specifically neuroinflammation, its symptoms and subsequent related neurodegenerative diseases. Here CBD, CBG and THC are highlighted as potent phytocannabinoids that combat the neurodegenerative effects of PCC. The scope is adequate and the narrative is appropriate. The problem of PCC is introduced concisely and is categorized into a range of mechanisms: oxidative stress, immunological alterations and inflammatory damage. This reviewer sees this as a good attempt at parsing through the research on PCC given how under-developed the field is and how variable the distribution of symptoms can be.

RESPONSE: We appreciate your valuable comment, and we share the idea that it is necessary to increase the study of PCC.

  1. To further include the latest research on the molecular structure of cannabinoids and strengthen the fit of the review to the scope of International Journal of Molecular Science, on the role of cannabinoids (section 5), this reviewer recommend referencing the conformational flexibility of CBG as a distinguishing factor that relates to its inhibitory character and pharmacodynamics profile: Bioactivity of the cannabigerol cannabinoid and its analogues–the role of 3-dimensional conformation. Organic & Biomolecular Chemistry, 2023.

RESPONSE: According to this paper, it is clear how different conformers play an important role for ligand-receptor recognition which influences pharmacological effects. In this review we are not considering the ligand-receptor recognition features, we are only focused in clinical advantages on PCC of some phytocannabinoids: cannabidiol (CBD), cannabigerol (CBG) and Delta-9-tetrahydrocannabinol (THC) studied elsewhere. But your comment is interesting and could generate a new review focusing on the structural flexibility of these molecules when interacting with receptors related to neuroinflammation, considering the aim of the interesting article that we consulted by your recommendation. We mention briefly about this point in the end of introduction and in the conclusion.

  1. Also, to illustrate the molecular structure of these discussed cannabinoids may be informative.

RESPONSE: We add figure 2.  We have included a 2D scheme of cannabidiol (CBD), cannabigerol (CBG) and Delta-9-tetrahydrocannabinol (THC) as the main importance for this review.

  1. On possible alternative therapies (section 6) it would be more useful to quantify the data from each study into the review (such as line 500 “can improve” by how much?). The aim here is that at least with the values present in each mention of each study, this section could be appended with a clearer idea of an otherwise foggy direction that this branch of research has. To summarise, based on the data amassed, how likely and to what extent are these alternative therapies contributing to the reduction in PCC inflammation?

RESPONSE: We have expanded the explanation of clinical trials showing how they measured or will measure the effect of Cannabis-derived drugs or compounds on symptoms in the post-COVID condition.  In these studies, validated scales or questionnaires were used to analyze and quantify the patient's perception of improvement in their symptomatology when Cannabis or its derivatives were administered. In relation to inflammation, none of these studies evaluated or will evaluate markers related to inflammation. However, we consider that according to the literature, the reduction of symptoms such as fatigue, anxiety, dyspnea or pain (to mention some of those that were evaluated in the studies) indirectly indicates that inflammation also decreased because in the literature are mentioned studies where is indicated a relationship between persistent symptomatology in long-COVID-19 with the inflammatory condition (for example in 10.1016/j.mad.2024.111915; 10.3390/life13112121; 10.1111/papr.13277).

  1. Line 156 repeated use of “can cause”

RESPONSE: We performed the suggested modification.

  1. Please confirm the originality of the 2 figures. If they are not original, proper references should be given and permission to use should be granted.

RESPONSE: All figures are originals designed by authors using the BIORENDER program paid by our team.

  1. A question for the authors that is maybe outside of the scope of this review, around line 535 there is the case study of a 60 year old man using cannabinoids to alleviate his symptoms. Is there a direct/quantitative way to measure and monitor PCC outside of its secondary symptoms (pain insomnia etc).

RESPONSE: Currently, there are still no direct markers related to PCC, or neuroinflammation. Magnetic resonance imaging has been used to demonstrate neuroinflammation in the CNS, however, it is difficult for patients to have access to it due to the cost involved. As described in the clinical trials, scales of psychological symptoms and alterations in cognition, memory and attention are evaluated, the latter are also indirect markers that tell us about neuroinflammation. In the case of PCC, instruments that evaluate quality of life are also used. However, we agree with you, studies are needed that generate direct markers, in blood, saliva, and urine that can help in the diagnosis and monitoring of both PCC and neuroinflammation.

  1. Is the relationship between curing these secondary symptoms and reduction of PCC neurological or psychological (a la placebo effect)?

RESPONSE: We consider that the clinical improvement of PCC symptoms implies the reduction of neuroinflammation but also contributes to the psycho-emotional aspect of the patient, since the patient can improve their neurochemistry by reducing depression, anxiety, stress, and this impacts the limitation of symptoms and in their perception.
